# High coverage COVID-19 mRNA vaccination rapidly controls SARS-CoV-2 transmission in long-term care facilities

Pablo M. De Salazar [1,7 ✉], Nicholas B. Link [2,3,4,5,7 ✉], Karuna Lamarca[6] & Mauricio Santillana[1,3,4,5]

### Abstract

**Background** Residents of Long-Term Care Facilities (LTCFs) represent a major share of COVID-19 deaths worldwide. Measuring the vaccine effectiveness among the most vulnerable in these settings is essential to monitor and improve mitigation strategies.

**Methods** We evaluate the early effect of the administration of BNT162b2-mRNA vaccine to individuals older than 64 years residing in LTCFs in Catalonia, Spain. We monitor all the SARS-CoV-2 documented infections and deaths among LTCFs residents once more than 70% of them were fully vaccinated (February–March 2021). We develop a modeling framework based on the relationship between community and LTCFs transmission during the pre-vaccination period (July–December 2020). We compute the total reduction in SARS-CoV-2 documented infections and deaths among residents of LTCFs over time, as well as the reduction in the detected transmission for all the LTCFs. We compare the true observations with the counterfactual predictions.

**Results** We estimate that once more than 70% of the LTCFs population are fully vaccinated, 74% (58–81%, 90% CI) of COVID-19 deaths and 75% (36–86%, 90% CI) of all expected documented infections among LTCFs residents are prevented. Further, detectable transmission among LTCFs residents is reduced up to 90% (76–93%, 90% CI) relative to that expected given transmission in the community.

**Conclusions** Our findings provide evidence that high-coverage vaccination is the most effective intervention to prevent SARS-CoV-2 transmission and death among LTCFs residents. Widespread vaccination could be a feasible avenue to control the COVID-19 pandemic conditional on key factors such as vaccine escape, roll out and coverage.

### Plain language summary

A large number of COVID-19 infections and deaths have occurred in older adults who reside in long-term care facilities (LTCFs). In this study, we determine whether high rates of vaccinations among older adult residents of LTCFs in Catalonia (Spain) reduced transmission of SARS-CoV-2 —the virus that causes COVID-19— and prevented infections and deaths after most vaccinations were complete. We use statistical modeling to compare the expected number of infections and deaths in LTCFs if vaccination would not have occurred to those actually observed after vaccinating more than 70% of residents. We estimate that 3 out of 4 SARS-CoV-2 infections and deaths were prevented due to vaccination, and that SARS-CoV-2 transmission in LTCFs was reduced by 90%. Our study shows that high-coverage vaccination is a highly effective way to prevent SARS-CoV-2 transmission and death among vulnerable populations

[1] Center for Communicable Disease Dynamics, Department of Epidemiology, Harvard TH Chan School of Public Health, Boston, USA. [2] Department of Biostatistics, Harvard TH Chan School of Public Health, Boston, USA. [3] Machine Intelligence Lab, Boston Children's Hospital, Boston, USA. [4] Computational Health Informatics Program, Boston Children's Hospital, Boston, USA. [5] Department of Pediatrics, Harvard Medical School, Harvard University, Boston, USA. [6] Home Hospitalization Unit, Department of Infectious Diseases, Dos de Maig Hospital, Universitat Autònoma de Barcelona, Barcelona, Spain. [7] These authors contributed equally: Pablo M. De Salazar, Nicholas B. Link. ✉email: pablom@hsph.harvard.edu; nicklink@g.harvard.edu

Widespread vaccination has the potential to reduce SARS-CoV-2 infections and deaths, and subsequently improve social and economic conditions[1]. Available mRNA COVID-19 vaccines have been approved due to their capacity to reduce symptomatic disease, hospitalizations, and deaths in clinical trials[2,3]; evidence of their real-world effectiveness is growing[4–6], but confirmation still remains limited to certain populations and settings[7–9].

Among all populations, residents of long-term care facilities (LTCFs) represent a major share of COVID-19 deaths, with a 7-fold higher incidence of death compared to the general population in the US[10,11]. As a consequence, they have been prioritized for vaccinations in most settings. While observational data post-vaccination are still under evaluation at the time of this work, particularly regarding residents of LTCFs[4,12], early assessments on whether clinical trial results are good indicators of vaccine effectiveness in LTCFs would help refine control strategies[1].

In this study, we aim to quantify the early effect of the administration of the BNT162b2 mRNA COVID-19 vaccine on reducing the risk of SARS-CoV-2 transmission and COVID-19 death among residents of LTCFs in Catalonia (Spain), where high rates of full vaccination among individuals older than 64 years (>90% coverage) were reached around 3 months after vaccinations began on December 27, 2020. Prior to the vaccination campaign, the control of SARS-CoV-2 transmission in LTCFs in Catalonia[13] relied on: (a) protocolized prevention measures at the individual- and facility-level, and (b) rigorous and timely case-ascertainment (including passive case and active case detection) and isolation standards. Protocols regulating the conditions of external visits to the facilities, as well as screening protocols for exits/entries of residents were tightened between December 2020 and January 2021[14] but were relaxed once the vaccination campaign was completed. Further, restrictions on individuals' mobility were implemented by the Catalonian Government at different levels through the territory based on epidemiological risk; in all of Spain, the tightest restrictions were applied in March–June 2020 (which included a country-wide shelter-in-place intervention leading to control of SARS-CoV-2 transmission) and in December 2020–January 2021[15].

We show that high vaccination coverage results in a substantial reduction in transmission amongst LTCFs' residents, preventing around 3 in 4 documented infections and COVID-19-related deaths. Our analyses provide evidence that vaccination is the most effective intervention in controlling SARS-CoV-2 spread to date among vulnerable populations.

## Methods

**Data, population, and variables.** The target population analyzed in this work was all individuals older than 64 years living in care homes in Catalonia, estimated to be around 58,000 in total (see details in Supplementary Methods) between July 2020 and March 2021.This population was vaccinated using the BNT162b2 mRNA COVID-19 vaccine following the guidelines of the Spanish Ministry of Health. The epidemiological data used in this study were collected from a publicly available repository provided by the Health Department depending on the Generalitat de Catalunya, the Government of Catalonia. Data on LTCF are collected and updated on a daily basis using health reports from the Primary Care Clinical Station (Care home census), Aggregated Care Health Register (PCR results and deaths), Orfeu, the program for registering virological test results in care homes, and the Catalan Shared Clinical Record where vaccination are registered.

We defined three COVID-19 outcomes to evaluate vaccine efficacy: (a) documented infections, comprised of all new infections reported during the study period, independent of symptoms and vaccination status, (b) documented deaths, comprised of all deaths attributable to COVID-19 reported during the study period, also independent of vaccination status, and (c) detected county-level transmission (herein detected transmission, used as a binary indicator of transmission) defined as at least one documented infection in any facility within a county per unit of time. Each case was assigned the date of diagnostic or testing as the reporting date. LTCFs' residents were defined as those living in a LTCF and older than 64 years. The general population, or "community", was defined as all people in a specific area not living in LTCFs. All vaccinated individuals received the two doses of the BNT162b2 mRNA vaccine. Documented COVID-19 infections in LTCFs over time, identified with a diagnostic test (PCR or antigen test), were assumed to capture most infections given the surveillance protocols in place. Beginning in July 2020, the ascertainment of cases in LTCFs in Catalonia included symptomatic surveillance, tight outbreak investigation and testing of all contacts upon identification of one single case, as well as regular screening of all individuals and staff of a facility independently of symptoms (see further details in Supplementary Methods). Documented deaths attributable to COVID-19 included those with laboratory confirmation and those meeting both clinical and epidemiologic criteria. We used three spatial resolutions for our analysis, determined by the level of aggregation in the data: (a) county level, which corresponds to the definition and boundaries of each "comarca" ($n = 41$); (b) healthcare-area level, which corresponds with the definition and boundaries of each "regió sanitaria" ($n = 9$); and (c) regional level, which refers to the largest spatial resolution corresponding to the whole Autonomous Community of Catalonia. For further details see Supplementary Methods.

**Counterfactual models and statistical analysis.** We generated multiple time series of daily confirmed infections, deaths, and vaccinations in LTCFs, aggregated at the healthcare-area level, and at the regional (highest aggregation) level. Similarly, we collected daily confirmed infections in the general population, at the healthcare-area level and regional level. For each of these time series, we took a moving weekly average around each day to smooth the daily variation of reporting. Further, we generated a time series of the detected county-level transmission by week. The pre-vaccination period was from July 6 to December 27, 2020, when vaccination in LTCFs began. We evaluated the impact of vaccines during an evaluation time period from February 6 to March 28, 2021, the subsequent time period after which 70% of residents were vaccinated with two doses. The 70% threshold was chosen to represent the estimated herd immunity— the estimated level of immunity in a population that prevents uncontrolled spread of infections[16]. As a sensitivity test, we evaluated the effect of partial vaccination in a longer period starting on January 14, 2021, when 70% of residents had received the first dose of the vaccine.

We built regression models to predict the number of infections and deaths among LTCFs' residents using community infections as inputs. These models were calibrated during the pre-vaccination period and then used to generate predictions in the absence of vaccines during the evaluation period. We compared the models' counterfactual predictions with observations; discrepancies were used to quantify the effects of the vaccine. Further, for each county with at least 1 pre-vaccination week with a transmission event in LTCFs and at least 1 week without one ($n = 36$), we built a logistic regression model to estimate the probability that at least one documented transmission would be observed among LTCFs' residents in a given week, using community infections as input. These models were trained on

the pre-vaccination period and then used to predict transmission events in the evaluation period among LTCFs' residents; the aggregated county-level predicted probabilities and the aggregated observed transmissions were compared to quantify vaccine effectiveness in LTCFs. See Supplementary Methods for further details on the models.

As sensitivity, due to the high zero-inflation of documented infections in LTCFs at the county level and the difficulty this provides for statistical modeling and inference, we also predicted the epidemic size at a higher aggregate level (i.e., healthcare area). For each area, we predicted LTCFs' infections and prediction intervals using model 1 and LTCFs' deaths using model 2.

## Results

### Predicting change in epidemic size and transmission events in all Catalonia.
We quantified the effect of administering the BNT162b2 vaccine: (a) on reducing deaths and documented infections among LTCFs' residents older than 64 years, and (b) on reducing detected transmission caused by SARS-CoV-2 in LTCFs in Catalonia.

Figure 1a shows the temporal evolution of infections documented in the community and in LTCFs between July 6, 2020 and March 28, 2021. For context, the cumulative vaccination coverage among all LTCFs' residents is shown in Fig.1b. Vaccination was deployed among residents and healthcare workers at similar times across LTCFs' facilities in the region, beginning December 27 and reaching more than 95% of 2-dose coverage within 2 months.

Figure 2a, b shows predictions and observations of documented infections and deaths in all of Catalonia over time. We estimated that between February 6 and March 28, 2021, vaccines prevented 75% of documented infections (36–86%, 90% CI) and 74% of deaths (58–81%, 90% CI). As well, our analysis shows that 2 weeks after 70% of residents were fully vaccinated, detected transmission was reduced by 69% (24–80% 90% CI), 54% (0–70%), 50% (0–68%), 69% (25–80%), and 90% (76–93% 90% CI) for each subsequent epidemiological week (Fig. 2c).

### Predicting change in epidemic size across healthcare areas.
Consistently, healthcare-area predictions showed lower-than-

expected documented infections and deaths at a more granular level for both target periods of analysis, as seen in Fig. 3 a, b, respectively. In 7 out of the 9 health areas, the documented infections during the analysis period were consistently lower than the expected infections, and in the two areas where observed infections were higher (Barcelona Ciutat and Metropolitana Nord, among the most heavily populated) they became lower than expected in the final 2 weeks. Also, infections in Alt Pirineu i Aran become lower than expected around 1–2 weeks earlier than vaccine interventions and remain low for the target time periods, which likely reflect strong specific lockdown measures implemented there. Further, observed deaths were higher during the early analysis period (~January) in 5 out of 9 healthcare areas (Barcelona ciutat, Camp de Tarragona, Catalunya Central, Girona, Metropolitana Sud), and later became significantly lower, consistent with the main analysis. Factors not considered in the model such as variation of mortality rates due to seasonality or spread of variants with higher lethality might have biased our estimates; in this case, the true number of prevented deaths and infections would be bigger than those estimated by our model. Supplementary Tables 1 and 2 summarize the number of documented infections and deaths averted by each healthcare area for the two target periods described in the "Methods". Note that because we used a linear model to predict deaths, some of the predictions are negative, which we truncated at zero. Due to the granularity of the area-level and the greater uncertainty around both models' predictions, many of the infection- and death-averted confidence intervals contain negative values, which would indicate an increase in either outcome. The locations with a greater population (Barcelona Ciutat and Metropolitana Nord) tend to have smaller confidence intervals because their greater population size yields more stable outcome values. While the healthcare area-level confidence intervals are wide, when we look at Catalonia as a whole (Fig. 2a, b) we see a more certain effect of vaccines.

## Discussion

In this study we showed that high vaccination coverage (over 70%) prevented around 3 out of 4 expected COVID-19 deaths among residents of LTCFs in subsequent weeks, which is

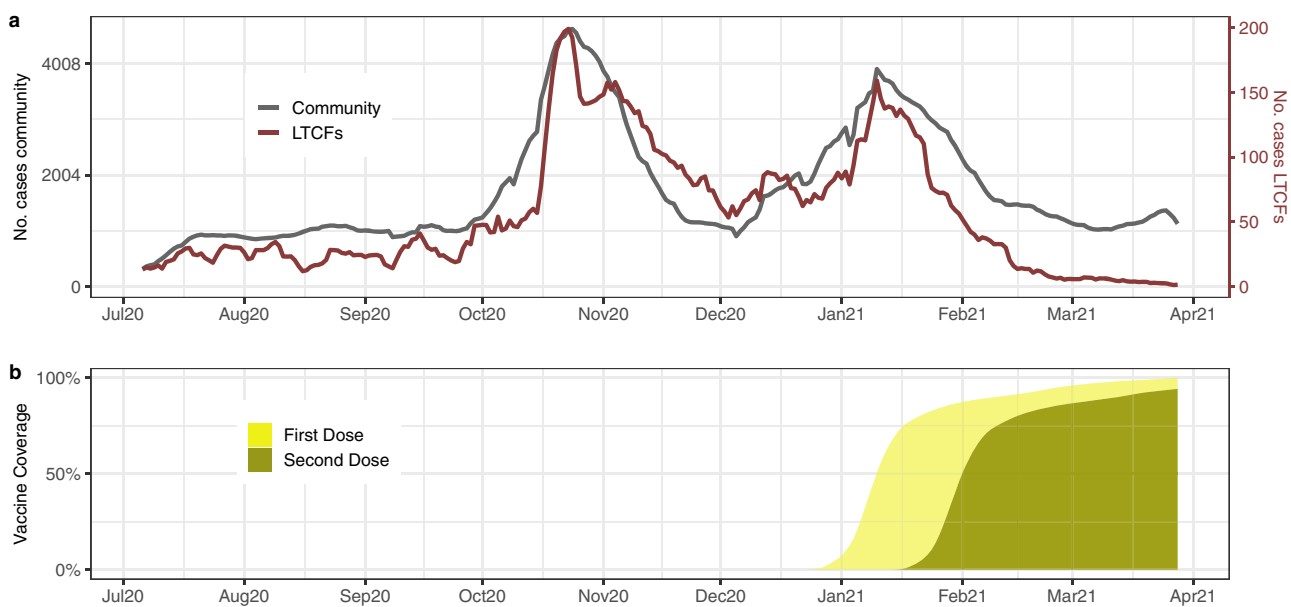

**Fig. 1 Documented infections and vaccinations in Catalonia, July 6, 2020–March 28, 2021. a** Comparison of the total community (gray) and LTCFs' documented infections (red) trajectories in Catalonia, Spain. **b** First and second dose vaccine coverage among LTCFs' residents.

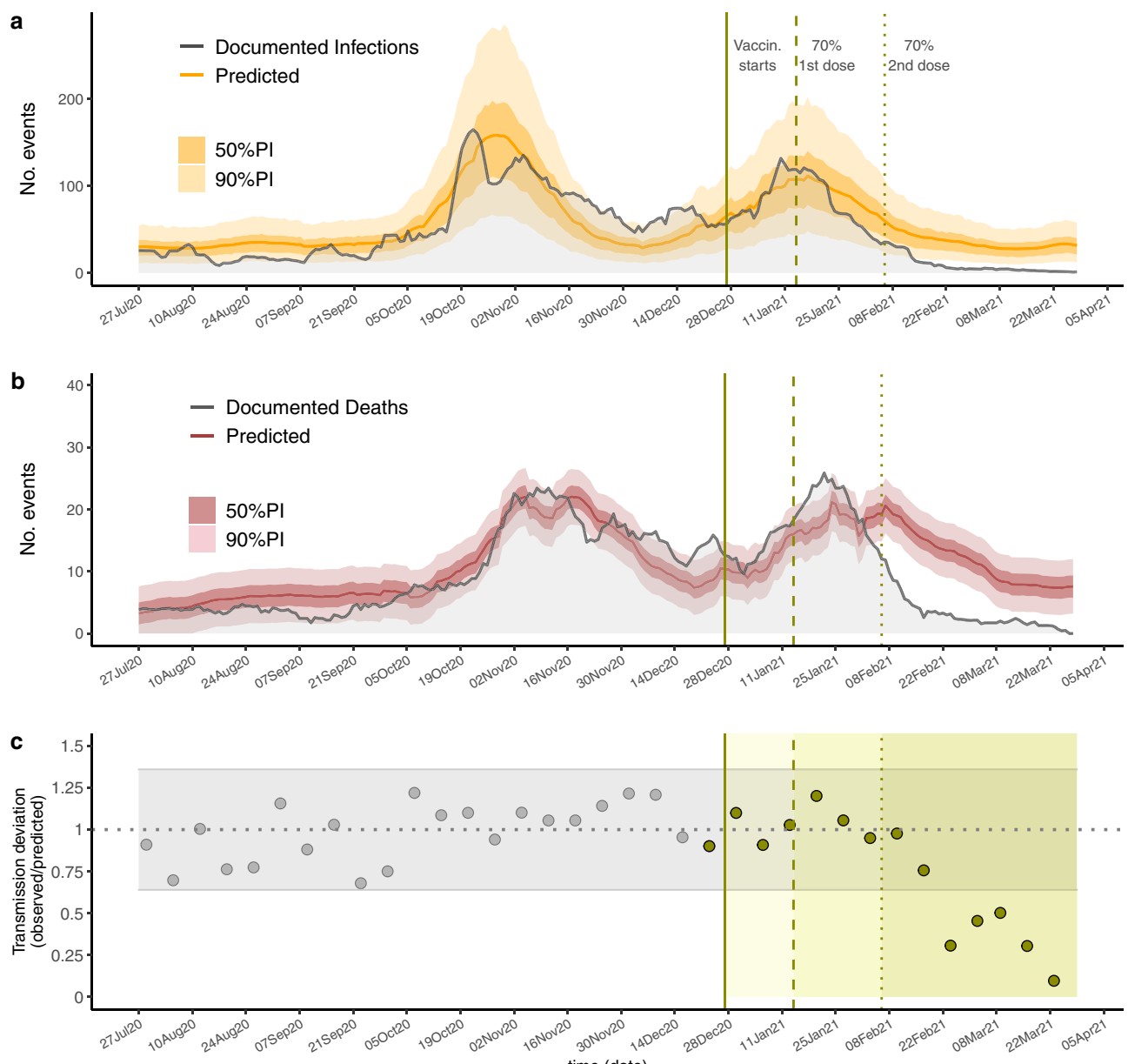

**Fig. 2 Predicted vs. observed SARS-CoV-2 infections, deaths, and transmission events in Catalonia.** The predictions for infections (**a**) and deaths (**b**) across all of Catalonia. The solid lines show the model predictions from training July 6, 2020 through December 27, 2020, the darker shaded background shows the 50% prediction intervals (PI), and the lighter background shows the 90% PI. Vertical lines show key analysis time points: when vaccination started (solid), when 70% of residents received the first dose and when 70% of residents received the second dose. **c** The ratio between observed and predicted transmission at county level in Catalonia, represented by point estimates, gray for the training period and green for the prediction period; gray horizontal ribbon represents the 90% confidence interval. Solid green areas represent the prediction periods after vaccination starts.

consistent with the vaccine effect on disease severity observed in clinical trials[2] and mortality reduction in other observational studies[6,12]. Further, we found a reduction in transmission after high vaccination coverage was reached while there still was transmission risk (spilled over from the mainly unvaccinated community transmission), which is caused by both a reduction in vaccinated individuals' probability of getting infected and a reduction in their probability of transmitting the virus[17].

LTCFs represent enclosed populations, where transmission is caused both by external introduction of the virus, mostly by the staff[18], and internal transmission. Observational findings from facilities with high infection-ascertainment, such as those studied here and elsewhere[4], may provide valuable information of what may be expected to happen in more general settings and

populations. Given this, we conclude that beyond the specific risk factors of the population in LTCFs, such as age or comorbidities, high-coverage vaccination can rapidly control SARS-CoV-2 transmission in an enclosed population. Particularly, our analysis suggests that transmission is reduced 3–10-fold at 1 month after the vaccination has reached 70% coverage. Of note, our estimates of infections and deaths do not differentiate between infections in vaccinated and unvaccinated individuals, and therefore can be interpreted as the population-level effect of vaccination.

Our methodological assumptions may lead to underestimation of the true vaccine effect. First, our definition of transmission events at the county-level does not capture the facility-level transmission; thus, our results may fail to capture more dramatic facility-level

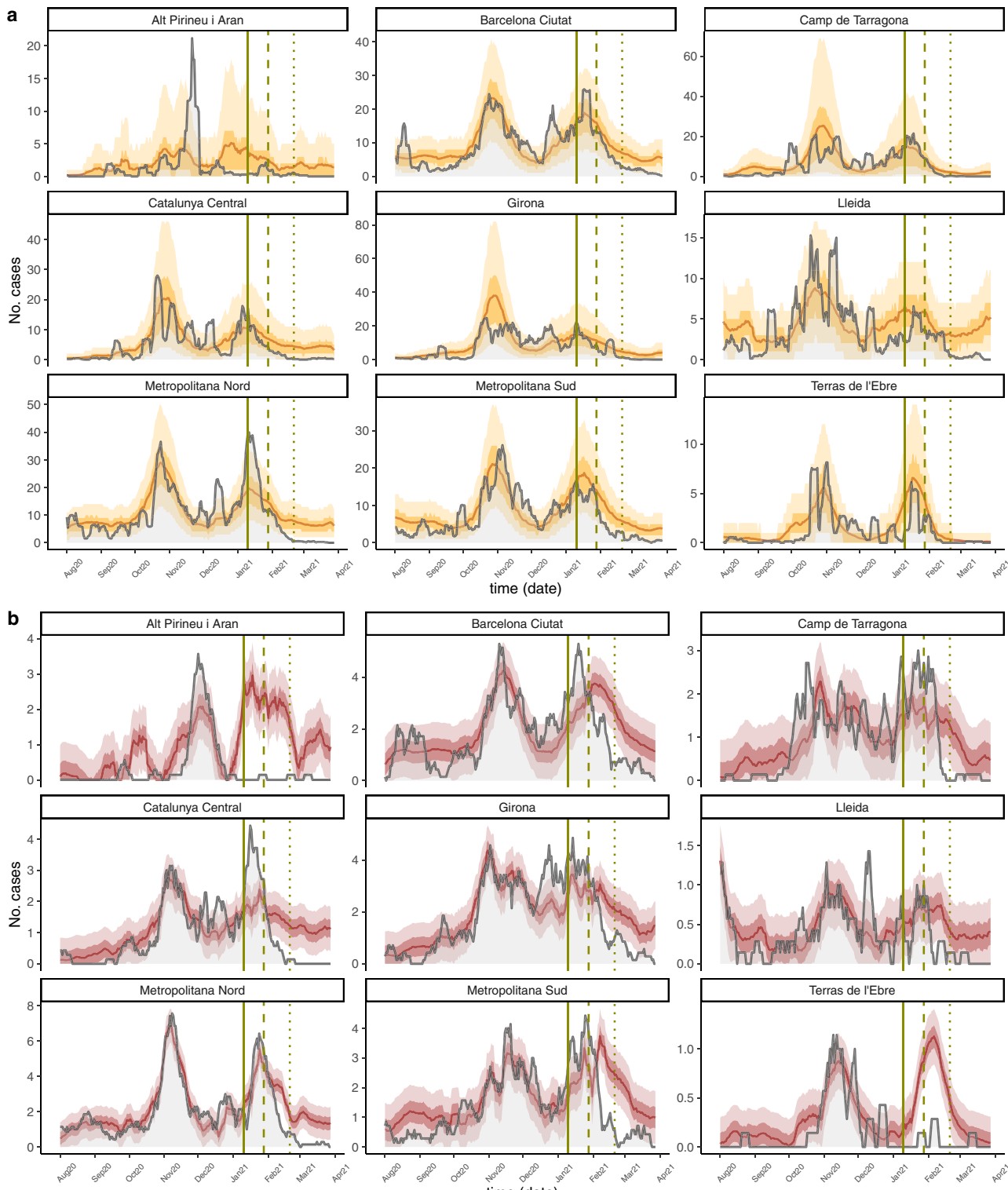

**Fig. 3 The epidemic size predictions for documented infections and deaths by healthcare area.** The gray lines are the observed documented infections (**a**) and deaths (**b**) and the yellow and red lines are the predicted infections, with ribbons for the IQR and 90% PI. Vertical lines show key analysis time points: when vaccination started (solid), when 70% of residents received the first dose and when 70% of residents received the second dose. Negative predictions are truncated and plotted as 0 for consistency.

reductions in transmission. Second, the counterfactual estimates of deaths produced by our models may be underestimations of mortality during the evaluation period. Indeed, the documented deaths were generally higher than those estimated by our models in December and January (Fig. 2b and Supplementary Fig. 2). It is

plausible that factors not considered in the model, such as seasonality[19] and/or the spread of more lethal variants[20] may have increased COVID-19 mortality in recent times. Further, as per guidelines, vaccinations in facilities with ongoing transmission were delayed, which would again lead to underestimation of the vaccine

effectiveness. On the other hand, it is possible that relaxing of case ascertainment after the vaccination campaign could lead to over-estimation of the effect on documented infections and detected transmission (but unlikely for documented deaths); therefore, we restricted our analysis to the period just after vaccination.

We acknowledge that our investigation has limitations. We assume that the rigorous screening standards in LTCFs in Catalonia led to infection ascertainment close to 100% and also that the ascertainment of community infections does not change dramatically over time. While this may appear unrealistic, public health authorities in Catalonia substantially increased the infection screening efforts in LTCFs before the vaccination campaign[14], beginning in December 2020.

Our model is based on the assumption that the time between disease transmission and case identification is not substantially different between individuals in the community and those in LTCFs. Further, regression models were not designed for accurate infection (or deaths) forecasting and as such, they may not fully capture the epidemiological dynamics, such as changes in COVID-19-restrictions policy over time. However, our efforts were focused on proper inference of the expected epidemic trajectory in the absence of vaccination. This goal is achieved as our models appear to reasonably capture the overall dynamics during the pre-vaccination time periods, even at high spatial granularity (Fig. 3).

Finally, there could be unmeasured confounders, such as behavior or policy changes, not captured by our models, that may have changed the dynamics of transmission between community and LTCFs over time—this may be the case in the health area Alt Pirineu i Aran prior to the beginning of the vaccination campaign. Nevertheless, tight restrictions on individuals' mobility for the whole territory were homogeneously implemented beginning December 2020 and during the period of analysis in this work.

In spite of these limitations, our analyses provide evidence that vaccination may be the most effective intervention available to date in controlling SARS-CoV-2 spread and subsequent risk of death. If our findings continue to be confirmed by future studies, then, conditional on important factors such as vaccine roll out, escape and coverage[21,22], widespread vaccination could be shown to be a feasible avenue to control the COVID-19 pandemic.

**Reporting summary**. Further information on research design is available in the Nature Research Reporting Summary linked to this article.

## Data availability
All data were obtained from a publicly available repository https://dadescovid.cat. Source data for all figures in the manuscript can be accessed as LTCF_vaccine_analysis/results at https://zenodo.org/record/5006849#.YNCbYy9h1QJ[23].

## Code availability
Code developed for the analysis can be accessed as LTCF_vaccine_analysis/code at https://zenodo.org/record/5006849#.YNCbYy9h1QJ[23].

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

## Acknowledgements
We thank Rebecca Kahn, Cristina Colls-Guerra, and Antoni Plasencia for their technical advice. P.M.D. was supported by National Institute of General Medical Sciences, grant number 5R35GM124715-02. N.L. was supported by the National Institute of Health Big Data Training Grant (T32 LM012411) and M.S. was partially funded by the National Institute of General Medical Sciences of the National Institutes of Health (R01 GM130668). M.S. and P.M.D. thank the Harvard Data Science Initiative for partially funding this work. The content is solely the responsibility of the authors and does not necessarily represent the official views of the National Institutes of Health.

## Author contributions
P.M.D., N.L., and M.S. conceived the study and designed and formulated the primary statistical models. P.M.D. and N.L. implemented the models, wrote the code and performed all of the analysis. P.M.D., N.L., K.L., and M.S. contributed to the inference model design, interpretation of results, and writing and editing for the manuscript. M.S. supervised the study.

## Competing interests
The authors declare no competing interests.
