## [Peer Review File. · Communications Medicine]

Reviewers' comments:

Reviewer #1 (Remarks to the Author):

An interesting analysis. I have two key concerns

1. The presentation, accounting for and discussion of other measures that were in place that could have altered dynamics
2. The choice of model for answering this question.

Methods

It would be helpful to have a description of the population being studied and the data streams used at the beginning of the methods.

I found the first sentence about the assumption of positive tests hard to interpret without knowing what the system/population these were being collected from was.

It would be helpful in the background to have any description of other control measures in place in the population and how these had changed in time also relative to the vaccination.

Please provide more justification of the model used. I would need more convincing that the logistic regression is the best way to model numbers of cases, and therefore I have concerns about how valid the inference can be from this model.

Results

What is the definition of “detected transmission” compared to “documented infections”? Please clarify this in the text.

Discussion

I think more should be explained about other control measures in the population. I don't think the “we suspect this is not the case” is not enough. This could also be compared to the first wave- what was happening that controlled the first wave.

Reviewer #2 (Remarks to the Author):

Ganna Rozhnova

University Medical Center Utrecht, The Netherlands

The authors evaluate the effect of rolling out BNT162b2 mRNA vaccines in LTHF on notified infections and deaths focusing on a well-defined case study (Catalonia in Spain). I think this study would provide a valuable and timely contribution to the current literature but it needs to be updated with respect to the most recent findings in other countries, comparison with other relevant studies, and discussion of the model assumptions. My specific suggestions on how the manuscript could be improved follow below. It would be useful if the revised manuscript included line numbers and highlighted the updated text.

Abstract:

I think the abstract should be clear without the need to read the full paper and currently it is not.

- At some point it is hard to follow whether the COVID-19 deaths and documented infections refer

to LTCF residents only or to the general population in Catalonia. Please make this clear by including “among LTCF residents”, “in the general population”, “in Catalonia” where necessary (e.g., “of all documents infections were prevented in Catalonia”).

- “Detectable transmission was reduced” in Catalonia overall?
- “Widespread vaccination could be a feasible avenue to control the COVID-19 pandemic.”

Depending on the speed of vaccination rollout and vaccination coverage in different age groups, vaccination on its own might not be sufficient to control the pandemic (Refs below). I would rephrase this sentence. Same concerns the discussion.

Scientific Advisory Group for Emergencies. Imperial College London: Unlocking roadmap scenarios for England, 18 February 2021; 2021. Available from:

<https://www.gov.uk/government/publications/imperial-college-london-unlocking-roadmap-scenarios-for-england-18-february-2021>.

Moore S, Hill EM, Tildesley MJ, Dyson L, Keeling MJ. Vaccination and non-pharmaceutical interventions for COVID-19: a mathematical modelling study. *The Lancet Infectious Diseases*. doi:10.1016/S1473-3099(21)00143-2.

João Viana, Christiaan van Dorp, Ana Nunes, Manuel Gomes, Michiel van Boven, Mirjam Kretzschmar, Marc Veldhoen, Ganna Rozhnova. Controlling the pandemic during the SARS-CoV-2 vaccination rollout: a modeling study, 24 March 2021, PREPRINT (Version 1) available at Research Square [<https://doi.org/10.21203/rs.3.rs-358417/v1>]

Introduction:

- Line 1: SARS_CoV-2 -> SARS-CoV-2
- Line 5: “but evidence of their real-world effectiveness remains limited [4,5]”. This sentence and references need to be updated. The evidence is growing that vaccines have high effectiveness not only in reducing severe disease and deaths but also asymptomatic infections (Refs below)

Moustsen-Helms IR, Emborg HD, Nielsen J, Nielsen KF, Krause TG, Molbak K, et al. Vaccine effectiveness after 1st and 2nd dose of the BNT162b2 mRNA Covid-19 Vaccine in long-term care facility residents and healthcare workers - a Danish cohort study. *medRxiv*. 2021;doi:10.1101/2021.03.08.21252200.

Chodick G, Tene L, Patalon T, Gazit S, Tov AB, Cohen D, et al. The effectiveness of the first dose of BNT162b2 vaccine in reducing SARS-CoV-2 infection 13-24 days after immunization: real-world evidence. *medRxiv*. 2021;doi:10.1101/2021.01.27.21250612.

Real-World Evidence Confirms High Effectiveness of Pfizer-BioNTech COVID-19 Vaccine and Profound Public Health Impact of Vaccination One Year After Pandemic Declared; 2021. Available from: <https://www.businesswire.com/news/home/20210311005482/en/>.

Hall, Victoria Jane and Foulkes, Sarah and Saei, Ayoub and Andrews, Nick and Oguti, Blanche and Charlett, Andre and Wellington, Edgar and Stowe, Julia and Gillson, Natalie and Atti, Ana and Islam, Jasmin and Karagiannis, Ioannis and Munro, Katie and Khawam, Jameel and Group, The SIREN Study and Chand, Meera A and Brown, Colin and Ramsay, Mary E and Bernal, Jamie Lopez and Hopkins, Susan. Effectiveness of BNT162b2 mRNA Vaccine Against Infection and COVID-19 Vaccine Coverage in Healthcare Workers in England, Multicentre Prospective Cohort Study (the SIREN Study); 2021.

Available from: <http://dx.doi.org/10.2139/ssrn.3790399>.

- “While observational data post-vaccination are scarce to date, particularly regarding LTCFs [8]”. I do not think this is true as the data on LTCF residents and workers have been the first to emerge. This sentence needs to be updated in light of most recent references and data (some are indicated above but the list is not complete).

Methods:

- It is not clear to which time period the study refers. For clarity, I would mention this first with further details supplied in the Appendix.

Results:

- One important assumption that I do not see addressed or discussed is that the authors seem to assume that there were no changes in COVID-19 related policy in Catalonia at all throughout the whole period of the study? I am not familiar with the local situation but it seems unlikely that this would be the case.
- I also did not understand how the authors accounted for breakthrough infections after vaccination that, to my knowledge, appear to be very common among vaccinated LTCF residents.
- The delay in reporting of notified infections should have decreased through the study period but I did not see this mentioned or discussed anywhere.

Discussion:

- First paragraph: please update the reference list.
- I found the discussion (and introduction) to be too short and not sufficient for understanding how exactly the author’s study and analyses are different from other relevant studies conducted in other countries such e.g. the UK, Denmark etc. It definitely needs to be updated and extended to give a more comprehensive comparison of the current results.
- Is the BNT162b2 mRNA vaccine the only vaccine that has been distributed during the study period in the LTCF?

Figures:

- Please add the year to x-axis tick labels.

Section S1: Typo: COVID019 -> COVID-19

Reviewer #3 (Remarks to the Author):

Catalonia started its mass immunisation programme on 27 December 2020. This study makes use of the corpus of data generated by the National Department of Health to evaluate the effect of two-dose vaccination on Covid-19 infections and mortality in nursing home residents in Catalonia.

The description of intervention is clear as well as the description of the objectives, and definition of variables. The methodology used is appropriate to respond to the research question. Findings are relevant and timely and provide guidance on the effects of vaccination campaigns in Long-Term care facilities in Catalonia.

Some comments:

-Consider whether people who gathered the data should be credited or provide permission for data use with a scientific goal.

-In the methods section, the definition healthcare care level, regional level and county level is confusing.

- Particularly I do not understand the difference of these two sentences which both mention the health care area level: 1) We generated multiple time series of daily confirmed infections (...) aggregated by *healthcare area level*. AND 2) Similarly, we collected daily confirmed infections in the general population, at the *healthcare area level* and regional level.

-What does "broader Catalonia region" mean and how is this relevant to the analyses? What does regional level mean, is this the county-level?

-A relevant confounder in the sensitivity analyses (using the period starting from Jan 14) is that the vaccination intervention began first in free-covid long term care facilities, while vaccination was delayed in those that had a case of covid. This may underestimate the effect of the vaccination campaign.

-Discussion, the primary outcome of clinical trial [2] was an episode of symptomatic Covid-19, rather than death.

Based on the responses and advice from reviewers, we have revised our manuscript. We feel that this process has improved the quality of our paper and has helped us present our results more clearly. Please find all our responses below.

Sincerely, on behalf of all authors,

Pablo M. de Salazar and Nick Link

Reviewers' comments:

Reviewer #1 (Remarks to the Author):

An interesting analysis. I have two key concerns

- 1. The presentation, accounting for and discussion of other measures that were in place that could have altered dynamics*
- 2. The choice of model for answering this question.*

We thank the reviewer for the helpful comments. We have addressed their concerns below.

Methods

1. It would be helpful to have a description of the population being studied and the data streams used at the beginning of the methods.

We have now included a description of the population and data streams at the beginning of the methods. Specifically, the following paragraph has been added to the manuscript:

“The target population analyzed in this work was all individuals older than 64y living in care homes in Catalonia, estimated to be around 58,000 in total (see details in supplementary information section S1), between July 2020 and March 2021. This population was vaccinated using the BNT162b2 mRNA COVID-19 vaccine following the guidelines of the Spanish Ministry of Health. The epidemiological data used in this study was obtained from --a publicly available repository provided by-- the Health Department depending on the Generalitat de Catalunya, the Government of Catalonia. Data on LTCF are collected and updated on a daily basis using health reports from the Primary Care Clinical Station (Care home census), Aggregated Care Health Register (PCR results and deaths), Orfeu, the program for registering virological test results in care homes, and the Catalan Shared Clinical Record where vaccination are registered.”

2. I found the first sentence about the assumption of positive tests hard to interpret without knowing what the system/population these were being collected from was.

Thank you. We have now extended and clarified the description of the assumption in the methods section as follows:

“Documented COVID-19 infections in LTCFs over time, identified with a molecular test (PCR or antigen test), were assumed to capture most infections given the surveillance protocols in place. Beginning in July 2020, the ascertainment of cases in LTCFs in Catalonia included symptomatic surveillance, tight outbreak investigation and testing of all contacts upon identification of one single case, as well as regular screening

of all individuals and staff of a facility independently of symptoms (see further details on supplementary information, Section S1). “

3.It would be helpful in the background to have any description of other control measures in place in the population and how these had changed in time also relative to the vaccination.

We have included an overall summary on the relevant control measures and their change over time in the background:

“In this study, we aimed to quantify the early effect of the administration of the BNT162b2 mRNA COVID-19 vaccine on reducing the risk of SARS-CoV-2 transmission and COVID-19 death among residents of LTCFs in Catalonia (Spain), where high rates of full vaccination among individuals older than 64y (>90% coverage) were reached around 3 months after vaccinations began in December 27, 2020. Prior to the vaccination campaign, the control of SARS-CoV-2 transmission in LTCFs in Catalonia [13] relied on: a) protocolized prevention measures at the individual and facility level, and b) rigorous and timely case ascertainment (including passive case and active case detection) and isolation standards. Protocols regulating the conditions of external visits to the facilities, as well as screening protocols for exits/entries of residents were tightened between December-January [14] but were relaxed once the vaccination campaign was completed. Further, restrictions on individuals’ mobility were implemented by the Catalanian Government at different levels through the territory based on epidemiological risk; in all of Spain, the tightest restrictions were applied in March-June 2020 (which included a country-wide shelter-in-place intervention leading to control of SARS-CoV-2 transmission) and in December 2020-January 2021 [15].”

4.Please provide more justification of the model used. I would need more convincing that the logistic regression is the best way to model numbers of cases, and therefore I have concerns about how valid the inference can be from this model.

Thank you for your clarifying question. First, we use three different models to study three different processes/outcomes: (1) the number of infections/cases in long term care homes, (2) the number of deaths in long term care homes, and (3) the probability of any infection occurring within any long term care home in a county. For (1), to model the number of infections in LTCFs we use a negative binomial regression, which is standard for modeling case count data. For (2), to model the number of deaths in LTCFs, we use a zero-intercept linear regression, since the relationship of deaths and cases appears to be monotonic and linear -and we assume no covid-19 attributable death would happen in the absence of infections. Model (3) is where we use the logistic regression model to estimate binary “transmission events” at the county level. We defined a transmission event if there was any reported infection in that county for that week (i.e. = 1 if there was at least one new infection and = 0 if there were no new infections). Since the outcome is binary and we wish to generate predictions of the probability of this outcome occurring, a logistic regression approach was chosen.

We analyzed this outcome since it is indicative of the efficacy of vaccines reducing SARS-CoV-2 transmission. This is because without a reduction in transmission, we would expect to see a reduction in the total number of infections proportional to the number of people vaccinated, but we wouldn’t expect to see a 100% reduction in infections. Using the binary outcome and the

logistic regression model allows us to analyze if there are more county-level 100% reductions in infections than we would expect. We hope this clarifies any concerns about the model

We have now included this information in section S1 of the Supplementary:

“Summarized description and justification of inference models:

“We use three different models to study three different processes/outcomes: (1) the number of infections/cases in long term care homes, (2) the number of deaths in long term care homes, and (3) the probability of any infection occurring within any long term care home in a county. For (1), to model the number of infections in LTCFs we use a negative binomial regression, which is standard for modeling case count data. For (2), to model the number of deaths in LTCFs, we use a zero-intercept linear regression, since the relationship of deaths and cases appears to be monotonic and linear -and we assume no covid-19 attributable death would happen in the absence of infections. Model (3) is where we use the logistic regression model to estimate binary “transmission events” at the county level. We defined a transmission event if there was any reported infection in that county for that week (i.e. = 1 if there was at least one new infection and = 0 if there were no new infections). Since the outcome is binary and we wish to generate predictions of the probability of this outcome occurring, a logistic regression approach was chosen.

We analyzed this outcome since it is indicative of the efficacy of vaccines reducing SARS-CoV-2 transmission. This is because without a reduction in transmission, we would expect to see a reduction in the total number of infections proportional to the number of people vaccinated, but we wouldn’t expect to see a 100% reduction in infections. Using the binary outcome and the logistic regression model allows us to analyze if there are more county-level 100% reductions in infections than we would expect. “

Results

5. What is the definition of “detected transmission” compared to “documented infections”? Please clarify this in the text.

Thank you. We have now specified the definition of our measured outcomes in the methods section, first paragraph:

“We defined three COVID-19 outcomes to evaluate vaccine efficacy: a) documented infections, comprised of all new infections reported during the study period, independent of symptoms and vaccination status, b) documented deaths, comprised of all deaths attributable to COVID-19 reported during the study period, also independent of vaccination status, and c) detected county-level transmission (herein detected transmission, used as a binary indicator of transmission) defined as at least one documented infection in any facility within a county per unit of time. Each case was assigned the date of diagnostic or testing as the reporting date. LTCFs residents were defined as those living in a LTCF and older than 64y. The general population, or “community”, was defined as all people in a specific area not living in LTCFs.”

Discussion

6. I think more should be explained about other control measures in the population. I don't think the "we suspect this is not the case" is not enough. This could also be compared to the first wave- what was happening that controlled the first wave.

Thank you. In addition to the new text included in the introduction (please refer to query #3) , we have rewritten the following paragraph in the discussion for conciseness:

"Finally, there could be unmeasured confounders, such as behavior or policy changes, not captured by our models, that may have changed the dynamics of transmission between the community and LTCFs over time --this may be the case in the health area Alt Pirineu i Aran prior to the beginning of the vaccination campaign (Supplementary Information Figure S1-S2). Nevertheless, tight restrictions on individuals' mobility for the whole territory were homogeneously implemented beginning December 2020 and during the period of analysis of this work."

Reviewer #2 (Remarks to the Author):

Ganna Rozhnova
University Medical Center Utrecht, The Netherlands

The authors evaluate the effect of rolling out BNT162b2 mRNA vaccines in LTHF on notified infections and deaths focusing on a well-defined case study (Catalonia in Spain). I think this study would provide a valuable and timely contribution to the current literature, but it needs to be updated with respect to the most recent findings in other countries, comparison with other relevant studies, and discussion of the model assumptions. My specific suggestions on how the manuscript could be improved follow below. It would be useful if the revised manuscript included line numbers and highlighted the updated text.

Abstract:

1. I think the abstract should be clear without the need to read the full paper and currently it is not.

Thank you. We have edited the abstract for clarity. It now reads as follows:

"Residents of Long-Term Care Facilities (LTCFs) represent a major share of COVID-19 deaths worldwide. Measuring the vaccine effectiveness among the most vulnerable in these settings is essential to monitor and improve mitigation strategies.

We evaluated the early effect of the administration of BNT162b2 mRNA vaccines to individuals older than 64 years residing in LTCFs in Catalonia, a region of Spain. We monitored all the SARS-CoV-2 documented infections and deaths among LTCFs residents from February 6th to March 28th, 2021, the subsequent time period after which 70% of them were fully vaccinated. We developed a modeling framework based on the relation between community and LTCFs transmission during the pre-vaccination period (July -December 2020) and compared the true observations with the counterfactual model predictions. As a measure of vaccine effectiveness, we computed the total reduction in SARS-CoV-2 documented infections and deaths among residents of LTCFs over time, as well as the reduction on the detected transmission for all the LTCFs.

We estimated that once more than 70% of the LTCFs population were fully vaccinated, 74% (58%-81%, 90% CI) of COVID-19 deaths and 75% (36%-86%, 90% CI) of all expected documented infections among

LTCFs residents were prevented. Further, detectable transmission among LTCFs residents was reduced up to 90% (76-93%, 90%CI) relative to that expected given transmission in the community.

Our findings provide evidence that high-coverage vaccination is the most effective intervention to prevent SARS-CoV-2 transmission and death among LTCFs residents. Conditional on key factors such as vaccine roll out, escape and coverage --across age groups--, widespread vaccination could be a feasible avenue to control the COVID-19 pandemic”

2. *At some point it is hard to follow whether the COVID-19 deaths and documented infections refer to LTCF residents only or to the general population in Catalonia. Please make this clear by including “among LTCF residents”, “in the general population”, “in Catalonia” where necessary (e.g., “of all documents infections were prevented in Catalonia”).*

Thank you. We have reviewed and clarified the text throughout our manuscript to address this valuable observation.

3. *“Detectable transmission was reduced” in Catalonia overall?*

Thank you for identifying this ambiguity in our manuscript. This statement refers to transmission among residents in LTCFs. We have corrected this statement as follows:

“Further, detectable transmission among LTCFs residents was reduced up to 90% (76-93%, 90%CI).”

• *“Widespread vaccination could be a feasible avenue to control the COVID-19 pandemic.” Depending on the speed of vaccination rollout and vaccination coverage in different age groups, vaccination on its own might not be sufficient to control the pandemic (Refs below). I would rephrase this sentence. Same concerns the discussion.*

Scientific Advisory Group for Emergencies. Imperial College London: *Unlocking roadmap scenarios for England*, 18 February 2021; 2021. Available from: <https://www.gov.uk/government/publications/imperial-college-london-unlocking-roadmap-scenarios-for-england-18-february-2021>.

Moore S, Hill EM, Tildesley MJ, Dyson L, Keeling MJ. *Vaccination and non-pharmaceutical interventions for COVID-19: a mathematical modelling study. The Lancet Infectious Diseases.* doi:10.1016/S1473-3099(21)00143-2.

João Viana, Christiaan van Dorp, Ana Nunes, Manuel Gomes, Michiel van Boven, Mirjam Kretzschmar, Marc Veldhoen, Ganna Rozhnova. *Controlling the pandemic during the SARS-CoV-2 vaccination rollout: a modeling study*, 24 March 2021, PREPRINT (Version 1) available at Research Square [<https://doi.org/10.21203/rs.3.rs-358417/v1>]

Thank you. We appreciate this important point. We have now modified our manuscript to clarify this. Specifically, in the abstract as follows:

Conditional on key factors such as vaccine roll out, escape and coverage --across age groups--, widespread vaccination could be a feasible avenue to control the COVID-19 pandemic.

In the discussion:

“If our findings continue to be confirmed by future studies, then, conditional on important factors such as vaccine roll out, escape and coverage [21,22], widespread vaccination could be shown to be a feasible avenue to control the COVID-19 pandemic.”

Introduction:

- Line 1: SARS_CoV-2 -> SARS-CoV-2

Corrected.

- Line 5: *“but evidence of their real-world effectiveness remains limited [4,5]”. This sentence and references need to be updated. The evidence is growing that vaccines have high effectiveness not only in reducing severe disease and deaths but also asymptomatic infections (Refs below)*

Moustsen-Helms IR, Emborg HD, Nielsen J, Nielsen KF, Krause TG, Molbak K, et al. Vaccine effectiveness after 1st and 2nd dose of the BNT162b2 mRNA Covid-19 Vaccine in long-term care facility residents and healthcare workers - a Danish cohort study. medRxiv. 2021;doi:10.1101/2021.03.08.21252200.

Chodick G, Tene L, Patalon T, Gazit S, Tov AB, Cohen D, et al. The effectiveness of the first dose of BNT162b2 vaccine in reducing SARS-CoV-2 infection 13-24 days after immunization: real-world evidence. medRxiv. 2021;doi:10.1101/2021.01.27.21250612.

Real-World Evidence Confirms High Effectiveness of Pfizer-BioNTech COVID-19 Vaccine and Profound Public Health Impact of Vaccination One Year After Pandemic Declared; 2021. Available from: <https://www.businesswire.com/news/home/20210311005482/en/>.

Hall, Victoria Jane and Foulkes, Sarah and Saei, Ayoub and Andrews, Nick and Oguti, Blanche and Charlett, Andre and Wellington, Edgar and Stowe, Julia and Gillson, Natalie and Atti, Ana and Islam, Jasmin and Karagiannis, Ioannis and Munro, Katie and Khawam, Jameel and Group, The SIREN Study and Chand, Meera A and Brown, Colin and Ramsay, Mary E and Bernal, Jamie Lopez and Hopkins, Susan. Effectiveness of BNT162b2 mRNA Vaccine Against Infection and COVID-19 Vaccine Coverage in Healthcare Workers in England, Multicentre Prospective Cohort Study (the SIREN Study); 2021. Available from: <http://dx.doi.org/10.2139/ssrn.3790399>.

Thank you. We have updated the text in our paper to better reflect this:

“Available mRNA COVID-19 vaccines have been approved due to their capacity to reduce symptomatic disease, hospitalizations, and deaths in clinical trials [2,3]; evidence of their real-world effectiveness is growing [4–6], but confirmation still remains limited to certain populations and settings [7–9].”

- *“While observational data post-vaccination are scarce to date, particularly regarding LTCFs [8]”. I do not think this is true as the data on LTCF residents and workers have been the first to emerge. This sentence needs to be updated in light of most recent references and data (some are indicated above but the list is not complete).*

Updated, consistent with previous comment:

“While observational data post-vaccination are still under evaluation at the time of this work, particularly regarding residents of LTCFs [4, 12], early assessments on whether clinical trial results are good indicators of vaccine effectiveness in LTCFs would help refine control strategies [1].”

Methods:

- *It is not clear to which time period the study refers. For clarity, I would mention this first with further details supplied in the Appendix.*

Thank you. We have now included the time of the baseline and evaluation period in the abstract and the full time period in the methods.

Abstract:

“ We monitored all the SARS-CoV-2 documented infections and deaths among LTCFs residents from February 6th to March 28th, 2021, the subsequent time period after which 70% of them were fully vaccinated. We developed a modeling framework based on the relation between community and LTCFs transmission during the pre-vaccination period (July to December 27, 2020)”

Methods:

“The target population analysed in this work was all individuals older than 64y living in care homes in Catalonia, estimated to be around 58,000 in total (see details in supplementary information section S1), between July 2020 and March 2021. This population was vaccinated using the BNT162b2 mRNA COVID-19 vaccine following the guidelines of the Spanish Ministry of Health.”

Results:

- *One important assumption that I do not see addressed or discussed is that the authors seem to assume that there were no changes in COVID-19 related policy in Catalonia at all throughout the whole period of the study? I am not familiar with the local situation, but it seems unlikely that this would be the case.*

Thank you. This point aligns well with two of reviewer 1's concerns, which we have addressed by providing information in the introduction, discussion, and more detail in the Supplementary (see response to reviewer 1's 3rd query and 6th query). Specifically, we have clarified this within the limitations (in the discussion section):

“Further, regression models were not designed for accurate infection (or deaths) forecasting and as such, they may not fully capture the epidemiological dynamics, such as changes on COVID-19 restrictions policy over time. However, our efforts were focused on proper inference of the expected epidemic trajectory in the absence of vaccination. This goal is achieved as our models appear to reasonably capture the overall dynamics during the pre-vaccination time periods, even at high spatial granularity (Supplementary Information Figure S1-S2). ”

- *I also did not understand how the authors accounted for breakthrough infections after vaccination that, to my knowledge, appear to be very common among vaccinated LTCF residents.*

In our analysis, our outcomes change if any infection or death is reported, regardless of the vaccination status of the infected (or dead) person (in fact, we do not have the data to conduct our analysis based on vaccination status). Hence our outcomes include all infections (infections in un-vaccinated individuals and potentially breakthrough infections). We discuss this (data-imposed) limitation in the discussion. We note that despite this limitation, our analysis allows us to evaluate the evolution of population-level transmission (under the listed assumptions). Recall that our (modeled) counterfactual aims at capturing the total infections that would be expected under no vaccination. Later in the analyzed time periods, over 90% of LTCF patients were vaccinated, yet we were still observing *some* infections (though much fewer than we would expect without vaccination, as our analysis showed), and likely many of these were breakthrough infections. Therefore, when we quantify the reduction in infections, deaths, and transmission events due to the vaccine, this implicitly accounts for breakthrough infections because these infections are included in our final infection numbers. In summary, the approach

estimates the expected reduction of cases independently if those are among vaccinated or unvaccinated individuals.

We have now clarified this matter in the methods:

“We defined three COVID-19 outcomes to evaluate vaccine efficacy: a) documented infections, comprised of all new infections reported during the study period, independent of symptoms and vaccination status, b) documented deaths, comprised of all deaths attributable to COVID-19 reported during the study period, also independent of vaccination status, and c) detected county-level transmission (herein detected transmission, used as an indicator of transmission) defined as at least one documented infection in any facility within a county per unit of time.”

And the discussion:

“Of note, our estimates of infections and deaths do not differentiate between infections in vaccinated or unvaccinated individuals, and therefore can be interpreted as the population-level effect of vaccination.”

- *The delay in reporting of notified infections should have decreased through the study period but I did not see this mentioned or discussed anywhere.*

Our analysis is conducted on data beginning in July 2020 (rather than March 2020 which is when infections started) to minimize the potential confounders on reporting pathways and delays that were experienced at the beginning of the pandemic. We chose July 2020 because systematic (and geographically homogeneous) case ascertainment procedures were formalized and implemented in Catalonia. Indeed, the date of report that is used in this database is obtained mainly from the testing procedure/diagnostic (PCR or antigen test). While it is expected that there is heterogeneity between the true infection date and the date of testing, we assume that this time “delay” is short (less or about a week) based on existing literature (please see Lauer et al. 2020 below) . Moreover, the conclusions of our study hold assuming that the time between infection and testing is not significantly different between community and LTCFs infections. Nevertheless, our models are capable of predicting cases on the pre vaccination period (we used out-of-sample validation) and therefore we believe the interpretation on the estimates holds. We have now clarify this in the methods:

“Each case was assigned the date of diagnostic or testing as the reporting date.”

As well as in the discussion as a limitation of our assumptions:

“Our model is based on the assumption that the time between disease transmission and case identification is not significantly different between individuals in the community and those in LTCFs

(...)

However, our efforts were focused on proper inference of the expected epidemic trajectory in the absence of vaccination. This goal is achieved as our models appear to reasonably capture the overall dynamics

during the pre-vaccination time periods, even at high spatial granularity (Supplementary Information Figure S1-S2).”

Reference: Lauer, S.A., Grantz, K.H., Bi, Q., Jones, F.K., Zheng, Q., Meredith, H.R., Azman, A.S., Reich, N.G. and Lessler, J., 2020. The incubation period of coronavirus disease 2019 (COVID-19) from publicly reported confirmed cases: estimation and application. *Annals of internal medicine*, 172(9), pp.577-582.

Discussion:

- First paragraph: please update the reference list.

Updated

• I found the discussion (and introduction) to be too short and not sufficient for understanding how exactly the author's study and analyses are different from other relevant studies conducted in other countries such e.g. the UK, Denmark etc. It definitely needs to be updated and extended to give a more comprehensive comparison of the current results.

Thank you. We have now updated and extended the introduction and discussion addressing the reviewers concerns and including the evidence arising from the suggested studies.

- Is the BNT162b2 mRNA vaccine the only vaccine that has been distributed during the study period in the LTCF?

Yes. Clarified in the methods section, first paragraph:

“This population was vaccinated using the BNT162b2 mRNA COVID-19 vaccine following the guidelines of the Spanish Ministry of Health.”

Figures:

- Please add the year to x-axis tick labels.

Thank you. We have now added the year to the labels.

Section S1: Typo: COVID019 -> COVID-19

Corrected.

Reviewer #3 (Remarks to the Author):

Catalonia started its mass immunisation programme on 27 December 2020. This study makes use of the corpus of data generated by the National Department of Health to evaluate the effect of two-dose vaccination on Covid-19 infections and mortality in nursing home residents in Catalonia.

The description of intervention is clear as well as the description of the objectives, and definition of variables. The methodology used is appropriate to respond to the research question. Findings are relevant and timely and provide guidance on the effects of vaccination campaigns in Long-Term care facilities in Catalonia.

Some comments:

-Consider whether people who gathered the data should be credited or provide permission for data use with a scientific goal.

Thank you. The data included for the analysis of this work was obtained from a publicly available repository. We now included following statement in the methods section, crediting the institution responsible in gathering the data and the data stream:

"The data was obtained from a publicly available repository provided by the Health Department depending on the Generalitat de Catalunya, the Government of Catalonia. Data on LTCF is collected and updated on a daily basis using health reports from the Primary Care Clinical Station (Care home census), Aggregated Care Health Register (PCR results and deaths), Orfeu, the program for registering virological test results in care homes, and the Catalan Shared Clinical Record where vaccination are registered."

*-In the methods section, the definition healthcare care level, regional level and county level is confusing. Particularly I do not understand the difference of these two sentences which both mention the health care area level: 1) We generated multiple time series of daily confirmed infections (...) aggregated by *healthcare area level*. AND 2) Similarly, we collected daily confirmed infections in the general population, at the *healthcare area level* and regional level.*

Thank you. We have now included the definition of these spatial resolutions in the methods, which are determined by the names of the areas of aggregation of the available data :

"We used 3 spatial resolutions for our analysis, determined by the level of aggregation in the data: a) county level, which corresponds to the definition and boundaries of each "comarca" (n= 41) b) health care area level, which corresponds with the definition and boundaries of each "regió sanitària" (n=9), and c) regional level, which refers to the largest spatial resolution corresponding to the whole Autonomous Community of Catalonia. For further details see Supplementary Information Section S1."

-What does "broader Catalonia region" mean and how is this relevant to the analyses? What does regional level mean, is this the county-level?

Thank you. We understand this concern. We have now clarified the definition, as stated in the previous point and modified the sentence:

"We generated multiple time series of daily confirmed infections, deaths, and vaccinations in LTCFs, aggregated by healthcare area level, and by the regional (highest aggregation) level."

-A relevant confounder in the sensitivity analyses (using the period starting from Jan 14) is that the vaccination intervention began first in free-covid long term care facilities, while vaccination was delayed in those that had a case of covid. This may underestimate the effect of the vaccination campaign.

Thank you. We agree with the reviewer that this is an important confounder. We have included this as a limitation in the discussion:

"Further, as per guidelines, vaccinations in facilities with ongoing transmission were delayed, which would again lead to underestimation of the vaccine effectiveness. On the other hand, it is possible that relaxing

of case ascertainment after the vaccination campaign could lead to overestimation of the effect on documented infections and detected transmission (but unlikely for documented deaths); therefore, we restricted our analysis to the period just after vaccination.”

-Discussion, the primary outcome of clinical trial [2] was an episode of symptomatic Covid-19, rather than death.

Thank you. We have corrected the statement (introduction):

“In this study we showed that high vaccination coverage (over 70%) prevented around 3 out of 4 expected COVID-19 deaths among residents of LTCFs in subsequent weeks, which is consistent with the vaccine effect on disease severity observed in clinical trials [2] and mortality reduction in other observational studies [6,12].”

REVIEWERS' COMMENTS:

Reviewer #1 (Remarks to the Author):

Thanks to the authors for their comprehensive responses and edits. I find the manuscript much clearer now.

Reviewer #2 (Remarks to the Author):

The authors have taken into consideration my suggestions in the revised manuscript.

Few notes:

- The abstract has become much clear but it seems too long.
- Ref 23 can be updated, the manuscript has been accepted for publication in Nature Communications.
- Please include years December-January [14] (Line 70).

Reviewer #3 (Remarks to the Author):

The authors have adequately addressed the comments and concerns I raised.